

# Surface ozone in the southern hemisphere: 20 years of data from a site with a unique setting in El Tololo, Chile

Julien G. Anet[1], Martin Steinbacher[1], Laura Gallardo[2,3], Patricio A. Velásquez Álvarez[4], Lukas Emmenegger[1], Brigitte Buchmann[1]

[1]Laboratory for Air Pollution / Environmental Technology, Swiss Federal Laboratories for Materials Science and Technology, Duebendorf, Switzerland
[2]Departamento de Geofísica de la Universidad de Chile, Blanco Encalada 2002, piso 4, Santiago, Chile
[3]Center for Climate and Resilience Research (CR2), Blanco Encalada 2002, Santiago, Chile
[4]Dirección Meteorológica de Chile, Av. Portales 3450, Estación Central, Santiago, Chile

*Correspondence to*: Julien G. Anet (julien.anet@empa.ch), Martin Steinbacher (martin.steinbacher@empa.ch)

**Abstract.** The knowledge of surface ozone mole fractions and their global distribution is of utmost importance due to the impact of ozone on human health and ecosystems, and the central role of ozone in controlling the oxidation capacity of the troposphere. The availability of long-term ozone records is far better in the northern than in the southern hemisphere, and recent analyses of the seven accessible records in the southern hemisphere have shown inconclusive trends. Since late 1995, surface ozone is measured in-situ at "El Tololo", a high-altitude (2200 m asl) and pristine station in Chile (30°S, 71°W). The dataset has been recently fully quality-controlled and reprocessed. This study presents the observed ozone trends and annual cycles and identifies key processes driving these patterns. From 1995 to 2010, an overall positive trend of ~0.7 ppb/decade is found. Strongest trends per season are observed in March and April. Highest mole fractions are observed in late spring (October) and show a strong correlation with ozone transported from the stratosphere down into the troposphere, as simulated with a model. Over the 20 years of observations, the springtime ozone maximum has shifted to earlier times in the year which, again, is strongly correlated with a temporal shift in the occurrence of the maximum of simulated stratospheric ozone transport at the site. We conclude that background ozone at El Tololo is mainly driven by stratospheric intrusions rather than photochemical production from anthropogenic and biogenic precursors. The major footprint of the sampled air masses is located over the Pacific Ocean. Therefore, due to the negligible influence of local processes, the ozone record also allows studying the influence of El Niño and La Niña episodes on background ozone levels in South America. In agreement with previous studies, we find that during La Niña conditions, ozone mole fractions reach higher levels than during El Niño conditions.

# 1 Introduction

Tropospheric ozone ($O_3$) is a key atmospheric compound that plays an important role in many respects: It acts as a greenhouse gas, which is contributing to radiative forcing of up to 21% relative to the radiative forcing induced by $CO_2$



(Myhre et al., 2013). Ozone has adverse effects on crop yields and on human health, being an irritating agent and triggering asthma and cardiovascular diseases (Reich and Amundson, 1985;Brook, 2002;Fiscus et al., 2005). Ozone is also a major source of hydroxyradicals and, thereby, influences the oxidative capacity of the atmosphere (Crutzen, 1971;Staehelin et al., 2001).

Various processes determine the amount of ozone in the troposphere: ozone is naturally produced by oxidation of methane, by reaction of oxygen with lightning-induced NO-production, as well as by photochemical formation in the presence of volatile organic compounds (VOCs), nitrogen oxides (NOx) and sunlight (Crutzen, 1971;Crutzen, 1973;Crutzen and Zimmermann, 1991;Winer et al., 1992;Derwent et al., 1998). Thus, changes in ozone precursor emissions – which are partly due to anthropogenic activities – considerably influence the tropospheric ozone burden. However, a straightforward

attribution of emission changes to ozone trends is challenging due to the highly non-linear photochemistry, different (VOC- and NOx-limited) ozone production regimes and also photochemical loss processes (Crutzen, 1971;Sillman and He, 2002). A significant part of tropospheric ozone origins from stratosphere-troposphere-transport (STT), also known as stratosphere-troposphere-exchange (STE), happening e.g. in tropopause folds (Holton et al., 1995;Škerlak, 2014;Škerlak et al., 2014;Lefohn and Cooper, 2015). The STE is not evenly distributed over the globe and hotspots of transport of stratospheric

ozone into the planetary boundary layer exist in the region of the Rocky Mountains, Tibetan Plateau, Andes (around 30°S), storm tracks, and Indian Ocean (Škerlak et al., 2014). Recent modelling studies postulate that the contribution from STE to the tropospheric ozone burden may be as high as 23 % of the net photochemical production (Stevenson et al., 2006;Sudo and Akimoto, 2007). This contribution may change in the future due to climate change and could lead to more than 20% STT increase (Collins et al., 2003;Hegglin and Shepherd, 2009;Neu et al., 2014).

Ozone sinks include catalytic destruction involving $HO_2$ and photolytic destruction; $O_3$ can also be removed from the atmosphere by dry deposition, wet scavenging and uptake by vegetation (Galbally, 1968;Stevenson et al., 2006).

The first ozone observations in the atmosphere were performed in the nineteenth century in Montsouris/Paris (Volz and Kley, 1988). However, regular and geographically distributed measurements have become more established only in the second half of the 20th century. Nowadays, surface ozone observations are widespread and data are available from various

data repositories such as the World Data Centre for Greenhouse Gases (WDCGG) of the Global Atmospheric Watch (GAW) programme of WMO, or from regional environmental agencies like the European Environment Agency (AirBase), the US Environmental Protection Agency (CASTNET, AQS) or the Acid Deposition Monitoring Network in East Asia (EANET). However, observations in the southern hemisphere in general, and in South America in particular, are very sparse (Sofen et al., 2016a;Sofen et al., 2016b).

Following Cooper et al. (2014), Oltmans et al. (2013) and Parrish et al. (2012), emissions of anthropogenic volatile organic compound and hydrocarbon emissions have led to a strong rise of ozone production in the last century. In fact, ozone has been generally increasing by up to 3 or 7 ppb/decade in the southern (SH) and northern (NH) hemisphere, respectively, averaged over different time spans (all between 1971 to 2011 and at least averaged over 10 years, see e.g. Table 1 of Cooper et al. (2014)). Thorough research has been undertaken to explain the difference in the trends between the two hemispheres. A





possible explanation of the more pronounced trend in the NH is i) higher precursor emissions than in the SH, and ii) relatively short lifetime of ozone and subsequent lack of transport into the SH. Moreover, trends in the NH are very different from location to location. Recent work raised the attention to the flattening of the positive trend in NH tropospheric ozone at certain sites, especially at those located in Europe or eastern Northern America (Cooper et al., 2014). At most stations, this

finding can be explained by decreasing nitrogen oxides ($NO_x = NO+NO_2$) emissions in the developed western countries. Such a levelling off is currently not observed in the western Unites States as $NO_x$ sources in upstream regions such as eastern Asia are still significantly increasing (Cooper et al., 2012). In contrast, the few SH ozone monitoring stations only partly recorded a flattening of the trend (Cooper et al., 2014). These ozone time series either show increasing positive trends (Oltmans et al., 2013;Thompson et al., 2014), or no significant change at all (Oltmans et al., 2013;Cooper et al., 2014). A

world-wide map of ozone trends interpolated from the existing surface ozone measurement stations is not yet available. Wespes et al. (2016) recently tried to map ozone mixing ratio trends in the lower troposphere by using remote sensing satellite data from 2008-2013. They showed that ozone mixing ratios in the lower troposphere were generally decreasing all over the southern hemisphere and in most parts of the northern hemisphere during this period. However, this trend cannot be generalized as polluted areas of the world still show significant positive ozone trends.

Tropospheric ozone records often show a pronounced seasonal cycle. While in polluted areas, a strong photochemically driven summer peak is observed, a spring peak with stratospheric influence dominates in most continental pristine regions (Wang et al., 1998a;Wang et al., 1998b;Monks, 2000). Stations in the marine boundary layer in the SH such as Cape Grim, Australia, rarely reveal a distinct spring peak, but rather show a summer peak due to $HO_2$ photochemistry (Ayers et al., 1992;Monks, 2000). Measurements in the SH free troposphere (e.g. La Quiaca, Argentina, 3459 m asl) show a spring

maximum (Barlasina et al., 2013), similar to the NH ozone time series from unpolluted stations. This latter work is in contrast to the finding of Monks (2000), who concluded that the spring phenomenon is primarily a NH feature. Wang et al. (1998a) state that this NH spring peak originates from a combination of ozone-rich stratospheric influx (February-April) and formation by local ozone chemistry (April-June). A recent analysis of the ozone seasonal cycle at northern midlatitudes revealed a shift of these ozone spring peak concentrations backwards by 3 to 6 days per decade (Parrish et al., 2013). They

conclude that this feature may be explained by changes in atmospheric dynamics, possibly combined with variations in the geographical distribution of the precursor emissions.

The GAW ozone network has a satisfactory station distribution over the NH. This is not the case for the SH, wherethe network is very sparse and additional surface ozone time series are needed to understand the global picture of ozone dynamics. This paper describes a recently quality controlled 20-year surface ozone dataset from "El Tololo", a mountain site

in Chile, South America, where ozone and a standard set of meteorological variables have been measured since 1995. Recently, the station has been equipped with a new ozone monitor and a CO/CO2/CH4 analyzer by Empa (Swiss Federal Laboratories for Materials Science and Technology).

El Tololo is currently the only GAW station in the SH above the marine boundary layer regularly submitting tropospheric ozone data to WDCGG. Therefore, the station provides highly valuable information on the ozone distribution in the





unpolluted atmosphere. Gallardo et al. (2000) and Rondanelli et al. (2002) published analyses of the early phase of the ozone record, pointing at particular characteristics of the ozone time series in connection with large-scale Hadley circulation, cut-off lows and deep troughs, or related to transport from the boundary layer.

The main objective of this study is to characterize the complete time series of this station which is likely representative of a
large domain of the background SH and to provide insight into the key processes driving the observed variability and trends. Section two gives an overview of the measurement station, and the instrumentation. In section three, the data series are presented and interpreted. Finally, we present our conclusions in the last section.

## 2 Station characterization

### 2.1 Location

The atmospheric monitoring station "El Tololo" (TLL) is located in the Coquimbo-region at 2200 m asl, 30°.17S, 70°.79W, 400 km north of Santiago de Chile, around 90 meters below the top of the mountain "Cerro Tololo". At Cerro Tololo, astronomical telescopes and instruments are operated as the Cerro Tololo Inter-American Observatory (CTIO) which belongs to the US National Optical Astronomy Observatory (NOAO). The Chilean Meteorological Service (Dirección
Meteorológica de Chile, DMC) runs the El Tololo station on the CTIO area. The distance from El Tololo to the next bigger cities is 50 km to the NW (La Serena/Coquimbo) with smaller towns nearby (Vicuña 20 km NE, Paiguano 30 km NE, Andacollo 30 km SW, Ovalle 60 km SW, see Fig. 1). Fifteen kilometers North of El Tololo, the Elqui-Valley, which is dominated by agricultural activity, is located in a W-E elongation. The population density of this region is low, accounting 17 inhabitants per km$^2$.

### 2.2 Climatology

Climate at TLL is classified as cool and arid. Between 1995 and 2015, the average temperature was 13.4°C (see Fig. S3 in the supplementary material) and in most years, less than 70 mm of rainfall was registered which can be classified as a "desert climate" (BWk) following Köppen climate classification of Geiger (1961) and Kottek et al. (2006). The wind measured at TLL mostly blows from the SSW sector during the summer months (December, January, and February, DJF) and from the
NNE sector during the remaining time of the year (see Fig. S4). However, the wind direction data does not necessarily represent the free atmosphere, as the local topography at the station partially obstructs advection from the NW to N sectors. Moreover, turbulent eddies downwind of the mountain top influence the measured wind direction. Kalthoff et al. (2002) described the mesoscale wind regimes affecting the area.

In order to identify the main origin of the air masses arriving at TLL, backward trajectory simulations from the FLEXTRA
model (e.g. Stohl et al., 1995), calculated at Empa, were used. The model uses wind fields from the European Centre for Medium-Range Weather Forecasts (ECMWF), and subsequent analysis locates the source of the trajectories in the Northwest



to South sector with some rare events from the North and Southeast (see Fig. 2). The influence of air parcels from the northeastern parts of South America is minor since the Andes are efficiently blocking any advection of air masses from this direction. Therefore, local pollution events from the greater Santiago de Chile region are more relevant than large-scale pollution events originating, e.g., from biomass burning in the Amazon region. 71% of the 10-day-trajectories start at an altitude between 0 and 5000 meters, and 10 % of the 10-day-trajectories originate from altitudes higher than 8000 meters (see Fig. S5 in the supplementary material). The origins of the trajectories follow clearly distinguishable seasonal patterns: during the summer months (DJF), most trajectories originate from south of the station and from the lower troposphere. During the winter months (JJA), more trajectories start north of the station, as the south-eastern pacific high shifts to more northern latitudes at that time of the year (e.g. Rahn and Garreaud, 2014). This also explains why trajectories from the upper troposphere are more frequent in JJA compared to DJF (+50%), following increased subsidence. The mean air trajectory length is highest during spring time (SON) and lowest during fall (MAM, see Fig. S6).

Being located in the subtropics, TLL is rarely affected by frontal or cyclonic systems. Nevertheless, during spring and summer time (SON, DJF), cut-off lows and troughs from higher latitude may reach subtropical regions, thus influencing the large-scale advection patterns at TLL (Rondanelli et al., 2002) on short time scales. This leads both to advection of polar air masses as well as upward-transport of marine boundary layer air potentially polluted by human activities in the nearby cities, possibly influencing the chemical composition of the air at TLL.

Apart from meteorological frontal systems, climatological patterns like the El Niño-Southern Oscillation (ENSO) do influence the large-scale origin of air masses arriving at TLL. In Fig. 3, the wind climatology and the change in the wind field during an exemplary strong El Niño (1997-1998) and La Niña (1988-1989) event are shown (ERA Interim, 700 hPa wind, climatology from 1979-2015, see Methods section for details).

The subtropical Pacific high determines subsidence in the Tololo area year-around interrupted occasionally by passing fronts or cut-off lows (Fuenzalida et al., 2005). It also drives a low-level jet (LLJ) along the west coast of South America, which peaks in intensity in spring (Garreaud and Muñoz, 2005; Muñoz and Garreaud, 2005). During El Niño (La Niña) years, the Pacific high becomes weaker (stronger), which leads to negative (positive) anomalies in subsidence and coastal southerly winds.

As mentioned earlier, ozone transport due to STE is an important factor of the tropospheric $O_3$ burden particularly in the remote SH. The ERA interim climatology shows a "hotspot" of downward transport of stratospheric, ozone-rich air masses above TLL (cf Fig. 2 in Škerlak et al., 2014), especially during austral spring and summer. This can most probably only be explained by gravity wave triggering when air parcels originating from the southern pacific region suddenly encounter a strong change in orographic height (Andean barrier, up to 6000 m asl). Moreover, the weakening of the subtropical jetstream in DJF favors additional wave breaking, triggering downward transport of ozone through tropopause folds. This potentially leads to a higher burden of tropospheric ozone in DJF. We will discuss this subject more thoroughly later in this work.



## 3 Data and Methods

### 3.1 Ozone data at El Tololo

In 1995, TLL has been equipped with an ozone photometer and a set of meteorological sensors. Ozone at TLL is measured by UV absorption with a Thermo Environmental Instruments Inc. TECO 49-003 analyzer. The station is equipped with an external ozonator which allows producing defined levels of ozone to conduct performance checks. Measurements are done continuously and data are recorded on a Campbell Scientific 21X data logger as 15-minute averages. Zero and span checks on multiple levels are done twice weekly and once monthly, respectively, to keep track of the background signal and the instrument response. Regularly, the operator visually inspects the recorded data for obvious anomalies.

The TECO 49-003 analyzer measures the UV light absorption in the Hartley band (220-310 nm) where ozone is a strong absorber. The optical bench is a dual cell device which is connected to a mercury lamp (245 nm) as light source. Alternately, one cell is flushed with ozone-free air while the other is simultaneously flushed with sample air. This allows a correction for changes in light intensity and potential interfering species. The TECO 49-003 has a sensitivity of ±1 ppb and a precision of 2 ppb. The response time is on the order of 20 seconds to reach 95% of the new signal (TEI, 1992). In September 2010, instrument performance of the analyzer was assessed during the SMN/WMO/GAW - 4th Tropospheric Ozone Analyzer Intercomparison at the Servicio Meteorológico Nacional, Observatorio Central de Buenos Aires, Buenos Aires, Argentina. The instrument passed all checks and a comparison with an ozone traveling standard of the World Calibration Centre for Surface Ozone (WCC-Empa) confirmed the validity of the instrument calibration (see WCC report at http://empa.ch/documents/56101/250799/2010_BsAs_RCC-O3.pdf).

In early 2013, the station has been equipped with an additional instrument measuring greenhouse gases (Picarro Inc. G2301 CRDS for $CO_2$/$CH_4$/CO and $H_2O$ analysis) and a refurbished ozone photometer (Thermo Scientific, TE49c) using the same measurement principle as the TECO 49, as the latter is reaching its end of life. The two independent ozone time series agree well. A small systematic offset has been observed which is most likely due to different inlet heights above ground for the two measurement systems. A short overview comparing the overlapping measurements of the two devices is given in the supplementary material (section S1, Fig. S1). Figure S7 shows the time series of meteorological parameters.

### 3.2 Ozone data used in this study

In-situ ozone data from other surface stations in the GAW network (K-Puszta (Hungary), Ushuaia (Argentina), Cape Point (South Africa), Cape Grim (Australia), and La Quiaca (Argentina), see https://gawsis.meteoswiss.ch for more details) were downloaded from the World Data Centre for Greenhouse Gases (WDCGG; http://ds.data.jma.go.jp/gmd/wdcgg/) and are used for comparison purposes.

In addition to the surface ozone measurements, ozone sondes are recording valuable information about the vertical ozone distribution in the atmosphere e.g. within the SHADOZ project since 1998 (Thompson et al., 2007). Frequent data is available from Ascension Island (United Kingdom), Suva (Fiji), Watukosek (Java), Natal (Brazil), La Réunion (France),



Pago Pago (American Samoa), San Cristobal (Ecuador), and Irene (South Africa), where sondes have been launched every 2 to 6 days. Additionally, ozone soundings from Easter Island (Rapa Nui, Chile) have been kindly provided by the DMC (pers. comm. P. Velázquez) as the long term data were not available from any data centre yet. The ozone sondes are all equipped with an Electrochemical Concentration Cell (ECC). According to Thompson (2003) the agreement between the sonde and

the ground-based measurements lies around 2-7%.

Model data from two sources is used in this work to study the atmospheric large-scale influences on the local measurements at TLL: i) a Stratosphere-Troposphere-Transport (STE) climatology from Škerlak et al. (2014) and ii) wind field climatologies from the ERA-Interim reanalysis. The STE climatology allows identifying the footprint of a potential ozone contribution from the stratosphere, while the ERA-Interim reanalysis is used to help understanding the effect of climatic

variability associated, for example, with ENSO.

### 3.3 Methods

Prior to the long-term trend analysis, data are rigorously screened to eliminate all data potentially influenced by local pollution. In a first step, values above 55 ppb or below 10 ppb are flagged and visually inspected for outliers, as those data points mostly arise from zero/span checks or local influences (Fig. S2a in the supplementary material). In a second step, a

further filtering is applied inspired by the well-established approach from Thoning et al. (1989) applied to the long-term $CO_2$ record at Mauna Loa, Hawaii. Adopting their method to the conditions at El Tololo, data points with ozone concentrations experiencing a change of more than 4 ppb from one hour to the next are excluded (Fig. S2b). In a third step, a polynomial fit is applied to the nocturnal data (23:00-06:00 LT) and data points exceeding twice the standard deviation of all data points of the nocturnal fit computed over one night are excluded (Fig. S2c & d). An example of the effects of this filtering can be seen

in the supplementary material, section S2. A final visual inspection is performed, in order to exclude any periods of sampling problems or local pollution events referenced in the station log books (see Table S1). As well, correction of "false negatives" flagged by the automatic filtering routine can be recovered. Only then, the 15-minute ambient air data is averaged to hourly data, hourly averages to daily data, and daily averages to monthly data. The filtering of the data excludes approximately 8.9 % of the total data – this number includes missing datapoints as well, which are due to e.g. instrument failures.

Trends are computed from filtered, de-seasonalised monthly averages. Deseasonalization is done using an additive model (Kendall and Stuart, 1983), separating the seasonal component and the trend from the time series. Significance is estimated by means of a two-sided Student t-Test at the 5% significance level except where explicitly noted. In order to make the analysis more robust, all-time (24 h), nighttime (22-04 LT) and daytime (11-17 LT) data is analyzed separately. To discern changes in the diurnal and annual cycle, seasonal and monthly means based on hourly data of two periods (1996-2000 and

2011-2015) are computed. Correlation tests are assessed with the Pearson's product moment correlation coefficient, which is following a t-distribution.

To discern changes in the annual cycle, daily data have also been investigated. The Huang-Hilbert transform technique was selected to decompose those daily data into intrinsic mode functions (IMF) with the use of ensemble empirical mode





decomposition (EEMD, Huang and Wu, 2008;Wu and Huang, 2009). The EEMD allows decomposing the time series into a residual trend and various oscillating signals representative of variations at seasonal, synoptic and other time scales. EEMD turned out to be particularly powerful for this time series analysis as it succeeds to mimic the asymmetric seasonal cycle peaking in October (see later) which is rather hard to match with sine-curve fitting.

Datasets of daily averages from the other GAW stations do not undergo the filtering process. However, in order to distinguish more easily the time at which seasonal maximum ozone mole fractions occur, a running mean with a window of 4 days is applied to the data, including those of TLL.

Ozone sounding data (see section 3.2) was cumulated per station in order to get annual cycles as follows: For each station, all soundings with valid data were temporally aligned, in order to reach data densities of as many days per year as possible,

thus creating a small climatology. Multiple values for the same day of the year were averaged. Stations with less than 70% data coverage in a given year were rejected. Similarly to the surface stations, the annual cycle is smoothed with a running mean (width of filter: soundings with visually homogeneous, regular seasonal cycle=10 days; soundings with visually irregular seasonal cycles [Macquarie island, Marambio, and Ushuaia]=20 days) at pressure levels of 1000 hPa, 900 hPa, 800 hPa, 700 hPa, 600 hPa and 500 hPa in order to compute the timing of the seasonal ozone maximum for several altitudes.

The dataset used for large-scale stratosphere-to-troposphere ozone transport studies (Škerlak et al., 2014) is based on the ERA-Interim reanalysis (ERAI) data from ECMWF (Simmons et al., 2007). Driven by the wind field of ERAI, kinematic trajectories are calculated using an 3-steps iterative Eulerian integration scheme (Sprenger and Wernli, 2015). Trajectories are started on a dense global grid and calculated for 24 hours, where only the ones crossing the tropopause are flagged. These flagged trajectories are extended 4 days backward and forward, and those with a residence time in the troposphere

shorter than 48 h are excluded from the climatology. Škerlak et al. (2014) estimated a transport of 6.52 * 10^11 kg of ozone per trajectory which is given by the size of the grid cell. The mass flux is then a simple multiplication of the number of trajectories per unit of time times the mass of ozone transported downwards through a certain model level surface. For example, mass fluxes around TLL (see Fig. S8 in the supplementary material) amount to 8-10 kg/(km$^2$ s); for comparison, the half-morning production of ozone over the whole city of Santiago de Chile on a summer day amounts to around 6680

kg/3 hours (Elshorbany et al., 2009).

## 4 Results and Discussion

The complete available ozone dataset at TLL from 1996 to 2015 is shown in Fig. 4a. There are only a few extended data gaps, which are all documented in the station log book (see supplementary material, Table S1). The overall data availability is 87%.



## 4.1 Trend analysis

Time series of the filtered deseasonalized monthly means is shown in Fig. 4b. A highly significant increase of 0.66 ppb/decade is found for the entire period from 1996-2015 (p-value 0.0008). The variability of the filtered deseasonalized monthly means is within ±8 ppb.

The deseasonalized ozone time series can be further decomposed using the Huang-Hilbert transform technique. By analyzing the residual, a flattening and a reversal of the trend are observed since 2008 and 2010, respectively (Fig. 4c). Up to September 2010, the EEMD calculation reveals a positive linear trend of 0.67 ppb/decade which is in accord to the linear fit. The EEMD calculation reverses after September 2010 resulting in a negative trend of -0.41 ppb/decade.

The reversal around 2010 is about in agreement with Oltmans et al. (2013) who find indications of a declining trend in the
SH at one station (Cape Grim, Australia) at least, taking into account data up to 2010. A quick analysis of the trends of deseasonalized monthly averages of the stations Cape Point (South Africa) and Cape Grim (Australia) reveal a flattening (Cape Grim) respectively a reversal of the trend (Cape Point) from 2011 on, which is qualitatively in agreement with our findings for El Tololo (not shown). This phenomenon is even more pronounced in the NH, as shown in Cooper et al. (2014): global ozone concentrations which have been rising for more than 20 years tend to level off especially in the eastern parts of
the US and in Europe. This is primarily due to large scale changes in $NO_x$ and VOC emissions.

Changes in regular patterns over time provide useful information to understand the underlying drivers and processes. Therefore, the mean seasonal cycles for two 5-year periods, one at the beginning of the measurements (1996-2000), and the other one in the recent past (2011-2015), were analyzed for potential differences. In Figure 5, the monthly means, with the upper 95th and the lower 5th percentiles including associated uncertainties are presented.

The two periods show a very similar annual cycle. However, there are subtle differences: Especially in austral fall (February-March), the 5[th] percentile, mean, and the 95[th] percentile increased remarkably from the first to the second period. Among the three curves, the 5[th] percentile shows the most persistent positive deviations from February up to June in the more recent period. During the remaining of the year, changes are minor except for October, where 2011-15 shows slightly lower values of $O_3$. The annual cycle and some of the changes over time are partly driven by the annual cycle of ozone STE mass flux (see
Fig. S9 in the supplementary material). Especially the shift of maximum from October to August (see later, Fig. 10) is represented by a drop of ozone mole fractions in October in Fig. 5. This can be explained by the modelled decrease of ozone mass flux in September and October and, conversely, increase of the mass flux in August (Fig. S9). However, the increase of ozone mole fractions from March to May cannot be explained by STE only, as at that time, ozone STE mass flux shows negative anomalies (Fig. S9).

The attentive reader may have realized that the maximum of STE mass flux (Fig. S9) and the maximum of ozone mixing ration (Fig. 5) are shifted by two months. We explain this delayed response of ozone mole fractions to STE by following mechanisms: i) a certain amount of time is needed to equally distribute ozone stemming from STE in the lower troposphere (e.g. titration of NOx & HOx residing in the atmosphere) in order to reach chemical equilibrium and ii) deep convection



underestimation as well as seasonal cycle uncertainties within the ERA Interim dataset (Škerlak et al., 2014) lead to doubts concerning the exact onset of ozone STE mass flux maximum around the cordillera.

Table 1 undermines the findings of Figure 5. Only austral fall shows significant changes both during day- and night-times. All other seasons show either decreasing (summer) or slight increasing (winter and spring) trends, which are, however,

insignificant. In general, differences between nighttime and daytime trends are very low which indicates that TLL is a very good background station with similar ozone levels under free tropospheric conditions and under planetary boundary layer (PBL) influence. Mean ozone mole fractions at TLL only vary between 32.5 and 31.0 ppb during day and night, while other stations located near greater cities (e.g. Eastern US, Bloomer et al. 2010) report up to 50 ppb peak differences between nighttime and daytime. This finding is most probably attributed not only to the remote location, far away from pollutant

sources, but also to the high altitude located above the PBL.

The significant increase of ozone in fall cannot be attributed to changes in STE only, which rather show a slight decrease over the same period of time (see Fig. S9). Factors influencing the trend in austral fall may most probably be an increase of biomass burning in and Southeast Asia (e.g. Shi and Yamaguchi, 2014;Verma et al., 2015) and subsequent eastward transport of ozone precursors, which are highest during the months from January to April. The prevailing westerly conditions

(see Fig. 3) exclude any sensitivity of ozone mole fractions at TLL to emissions on the South American continent. Rather, the ozone increase may originate from regional pollution from the La Serena region, which – in fall – may get transported upwards due to the PBL height and occasional support by frontal systems. This latter assumption remains, however, hypothetical (see 4.3). A confirmation would require high-resolution numerical simulations to resolve the transport in the mountainous terrain of TLL.

Long-term trends similar to TLL can also be found at other stations in both the SH and the NH. We have made use of time series from Cape Point, Mace Head and from western US regions (e.g. Yellowstone NP, Lassen, and others), and most of them show a distinct increase in ozone mixing ratios up to the millennial years though mostly larger than at TTL, before showing signs of levelling off (GAW, 2013;Carslaw, 2005;Derwent et al., 2007;Derwent et al., 2013;Baylon et al., 2015;Cooper et al., 2014). TLL is distinct in that the reversal happens rather late, namely after 2010. However, some western

US stations still show signs of growing ozone concentrations, as they are downwind of pollutants from eastern Asia. Those stations, sampling air with origin over the Pacific Ocean, may be partly comparable with the TLL station which is as well sampling air from oceanic origin. Furthermore, the emissions of pollutants upwind of the western US coast are by far much higher than the emissions at similar longitudes from southern hemispheric regions. They are moreover prevalent over the whole year (industrial, coal mining and energy production being the greatest sources), compared to those of the SH which

are rather to be classified as seasonal peaks from biomass burning. Yet, as the effect of upwind pollutant release is visible in the datasets of the west coast US stations, it is well possible that long-term background ozone trends at TLL are also partly driven by pollutants originating from Southeast Asia.

Next, the ozone data from TLL will be discussed jointly with data from other stations before the influence of large-scale phenomena is discussed in chapter 4.3.





### 4.2 Annual cycles of O₃ at El Tololo, other ground based sites and from ozone sondes

In the following, the ozone data from TLL and other remote sites will be discussed with respect to the time of the ozone maximum during the year as well as the shape and the amplitude of the annual cycle.

Next to TLL, different monitoring stations in the NH (Jungfraujoch [JFJ], K-Puszta [KPS], Payerne [PAY], and Vindeln
[VDL]) and in the SH (Arrival Heights [ARH], Baring Head [BHD], Cape Point [CPT], Cape Grim [CGO], Ushuaia [USH], and La Quiaca [LQO]) were analyzed for comparison. Typical averaged annual cycles of selected available data (reduced to 4 typical cycles with KPS, summer maximum, TLL and LQO, spring maximum and CPT, winter maximum, for more clarity) are illustrated in Fig. 7 a) and b), where the annual cycle of NH stations has been shifted by 182 days for comparability purposes.

In order to characterize TLL as a certain type of station, a clustering of the ten stations into characteristic annual cycle categories has first to be done. All investigated ground-based in-situ ozone measurement either show a maximum in winter, spring or summer, mainly for the three following reasons: i) winter maxima can be mainly seen in clean, marine environments which are primarily driven by ozone depletion due to negative NOx-anomalies and methyl iodide production in summer (Combrink et al. (1995) or Nzotungicimpaye et al. (2014)). ii) Spring maxima are mainly influenced by STE of
ozone-rich air, influencing the regional chemical composition of air through dynamic forcing. iii) Summer maxima are mainly observed at stations influenced by ozone precursor emissions where photochemical production of ozone is the major process driving the annual ozone cycle.

TLL stands as a good example of a station featuring a spring maximum. An in-depth analysis of the drivers for this maximum is given below.

Photochemical ozone production is mainly following the sine-shaped availability of solar radiation, unless there is a strong seasonal variation in the precursors, e.g. due to biomass burning emissions. Dynamic processes such as the north – south movement of the ITCZ, shifts in synoptic weather patterns, and ozone entrainment by STE can result in less regular patterns, as the time of occurrence of the processes is usually concentrated over a shorter time period. The STE-effect at TLL, visible in Fig. S9, smoothly starts in June, reaches a peak in August, and regresses until December, staying at low levels until May.
Little to no STE influence is to be expected during the five months between January to May. This is the reason why ozone concentrations at TLL follow a slightly asymmetric course over the year. Hence, in order to understand the annual cycles of ozone "in three dimensions" at different latitudes in the SH, additional data is needed.

Therefore, ozone soundings from 12 SH remote locations (San Cristobal, Natal, Java, Ascension Island, Samoa, Fiji, La Réunion, Irene, Rapa Nui, Macquarie Islands, Ushuaia, and Marambio) have been analyzed. In Fig. 8, a compilation of our
analysis of both soundings and ground measurements is illustrated on an x-y-diagram, where the x-axis is the latitude of the sounding/station, and the y-axis represents the "peak-to-peak"-amplitude of the annual cycle. The size of the circle reflects the altitude of the measurement above sea level, and the intensity of the blue color the timing of the annual ozone maximum.





Ozone maxima are generally shifting to later times in the year with increasing altitude, i.e. peaks in the annual cycle at 500hPa occur later than the peaks in the annual cycle at 1000 hPa (see Fig. 8). This can to a great extent be explained by the location of the soundings, which are all launched in a marine environment. As we have learned before, in most marine surface ozone time series, a winter maximum prevails (around day 180-220). An ascending sounding will first sample air

that is influenced by photochemistry in the marine boundary layer. Thus, the higher a station is above sea level, the larger is the share of stratospheric input of ozone. At high altitudes, peak concentrations of ozone are therefore shifted towards later in the year. However, this picture is perturbed by the fact that i) north of 25°S, little to no STE occurs, and ii) the solar cycle is weak north of 10°S. There, interhemispheric mixing explains the late maximum. This process allows some of the NH pollutants to penetrate into the SH across the intertropical convergence zone (ITCZ). The ITCZ is also located in a region

where biomass burning prevails all-year long, leading to a very efficient upward transport of pollutants up to the tropopause. Following the position of the ITCZ, the most intense ozone production via the $NO_x$-$HO_x$-VOC cycle occurs late in the year. Therefore, we would expect that the maximum shifts from a summer maximum at the equator to a spring maximum at high latitudes.

This hypothesis is further supported by a majority of the soundings (see Fig. 8). Especially for soundings made from 25°S

southward down to the polar regions, a clear gradient from late maxima to earlier maxima is recognizable (shift from darker to brighter blues). This observation applies not only to soundings, but also to ground-based measurements. A trend to earlier maxima is observed from LQO to ARH, although a smoother grading (more stations) would be beneficial to solidify our hypothesis. When classifying TLL, which is reaching maximum values around mid-October, it can be noticed that the timing is a bit later than one would expect from extrapolating the ozone sonde measurements, as it is a continental station.

Maximum concentrations at CPT, CGO, BRH, USH or ARH are reached far earlier, mainly due to i) the marine influence at the stations and ii) the lack of stratospheric influence down to the surface.

The annual cycle at TLL can best be compared to sonde data taken at similar latitude and height. This is fulfilled best by RNO and FJI. A far weaker peak to peak amplitude is found at the surface station TTL (see Fig. 8) than in the free atmosphere. Part of this difference can be explained by the origin of air masses, which is dynamically driven. Focusing on

the soundings at Rapa Nui (RAP), during the summer and fall months, a strong high-pressure system with center over the island limits the advection of pollutants from the west, hindering photochemical production of ozone. Later in the season, the high pressure system moves slightly eastward, allowing transport of air masses from the Northwest (Oceania) towards the Southern Pacific and, therefore, advecting biomass burning pollutants from Southeast Asia and Oceania via the zonal wind field. In contrast, the sampled air mass at TLL is to a greater part pristine (see above), preventing strong photolytic ozone

production in summer.

## 4.3 Large-scale influences at TLL

The uniqueness of the seasonal cycle at TLL is linked to large scale dynamics and subsequent cycles in STE. While it is difficult to discern the "signature" of stratospheric ozone from tropospheric ozone, Figs. 9 and 10 provide some indication





that STE and associated ozone entrainment is a key driver for the observed ozone variability and trend at TLL. Fig. 9 shows the annual cycle of ozone concentration and the transport of air masses from the stratosphere. The latter are numbers averaged over 18 years and are extracted from the Škerlak et al. (2014) climatology. The two parameters show a strikingly similar patter, indicating that, in fact, STE may be a strong driver for $O_3$. Another indication for the coupling of $O_3$

concentration and STE is a coherent shift in the maximum of these quantities over the observation period towards an earlier occurrence in the year. This is illustrated in Fig. 10. For calculation, a 4-year sliding window of daily data was defined and run over all data between 1996 and 2015. Then, an empirical mode decomposition was done (Huang and Wu, 2008;Wu and Huang, 2009). Out of the Hilbert periodogram, the IMF resembling the most to an annual cycle is selected and the IMF-datapoints are extracted. The latter are averaged to get an average of IMF over the 4-year window. Finally, the day-of-year

matching the maximum value of the IMF is extracted. For the ozone time series, a regression of -10 days per decade was calculated. For STT, an even larger trend of -11 or -21 days per decade was obtained for the maximum number of trajectories of stratospheric origin and for the mass flux into the PBL, respectively.

Again, this evaluation of yearly maxima shows a strong resemblance between ozone volume mixing ratio and stratospheric influences for both STE-trajectories and mass fluxes into the PBL. Overall, we conclude that there are strong indications that

the annual cycle at TLL is to a major part driven by STE.

The shift in the seasonal cycle has already been presented in other studies for other locations (Parrish et al., 2013;Lin et al., 2014). For instance, spring peaks are observed in the NH to regress with a rate of 3 to 14 days per decade (Parrish et al., 2013). Parrish et al. (2013) also suggest that the relative contribution from the stratosphere may at least partly explain the shift in the annual cycle at high-altitude stations in the NH like Jungfraujoch, being located at 3580 meters asl. Yet, a

conclusive explanation for this shift of the seasonal cycle remains missing. Schnell et al. (2016) recently suggested that future climate change will shift the maximum of the ozone seasonal cycle to earlier in the year, but they did not provide any clear explanation for this phenomenon.

As TLL does represent a special case, it is of interest to also look at other modes of O3 variability. We will first summarize short-term variation patterns. Then, the dependence of O3 on large-scale, long-term oscillating patterns like the El Niño

southern oscillation will be discussed.

Concerning short-term variations, it is known from previous studies that a (anti)correlation between ozone and relative humidity exists at TLL, but only in very specific cases. Gallardo et al. (2000), analyzing the first years of data collected at Tololo, found such an anticorrelation between ozone and water vapor in summer in connection with upslope transport of boundary layer air associated with a thermally driven circulation. Rondanelli et al. (2002) investigated the effect of troughs

associated with a frontal zone passing over TLL, and could classify their observations in two categories: wet and dry events. During wet events, relatively humid air from the PBL is advected to TLL, and shortly after regression of relative humidity, ozone is rising rapidly. During dry events, ozone is rising, but relative humidity stays at normal, dry levels or drops even further. This study partly confirmed our finding of 4.1, where we explain the seasonal cycle with partial PBL influence. Carbon monoxide, a good PBL pollutant and hence an optimal tracer, has been measured in TLL since April 2013.





Therefore, the dependence of CO and ozone was investigated. This analysis revealed a significant correlation (not shown) in rare, specific episodes, during which less pristine air from the PBL – originating from the La Serena, Valparaiso and Santiago regions – is reaching TLL. Those events were not always associated with low potential vorticity values (PV streamer, reconstructed from ERA Interim data, not shown) or frontal zones, but some of them were. This confirms the finding of Rutllant et al. (2013), who, during the VOCALS-REx-campaign, found a persistent, regular South-Westerly advection via thermals, being able to transport air masses in the afternoon from the marine region into the Andes which would allow inbound transport of slightly more polluted air masses to TLL. An in-depth analysis of CO-$O_3$-correlations over several years would go beyond the scope of the paper, however would provide more robust conclusions.

On longer time-scales, also ENSO is known to dramatically influence climate variables in entire South America with a frequency of 3-5 years. This is the amount of time needed by the global climate system to switch from an El Niño phase to a La Niña phase. Although the ozone anomalies at TLL and the ENSO-index do not show any significant correlation, annual cycles of ozone during El Niño years and La Niña years show significantly different values especially in austral fall and spring (Fig. 11). During La Niña events, ozone levels reach higher values especially from September to November than during El Niño events. During La Niña years the Pacific high is stronger, which leads to more subsidence allowing the downward mixing of ozone rich air from the upper troposphere possibly connected with STE processes. On the contrary, El Niño years result in a weaker Pacific high and in diminished subsidence. Moreover, based on Fig. 4 in section 2.1, the strengthening of the low-level southerly flow during La Niña events may allow air parcels from the greater Santiago area to reach TLL more easily than during El Niño years. Also, cooler conditions during La Niña years may allow transport of PAN over larger distances. PAN can act as a reservoir for organic nitrates and releases nitrogen oxides after thermal decomposition which can lead to an efficient production of ozone in pristine environments such as TLL (see e.g. Fischer et al. (2014)). Lastly, large-scale anomalies of meteorological conditions over the Pacific could be the major driver of the differences during El Niño and La Niña conditions. As shown in the modelling studies of Doherty et al. (2006) and Sekiya and Sudo (2012), decrease in total column ozone is found in the eastern Pacific region during El Niño conditions. They attribute that to a decrease in NOx-production due to less lightning. It is highly probable that this has an effect on the ozone cycles of TLL illustrated in Fig. 11 also. Currently, the lower NOx-values in the TLL region during El Niño events cannot be confirmed due to missing measurements.

## 5 Conclusions

The 20 year-long surface ozone time series of El Tololo, Chile (TLL) has been presented and analyzed. It was characterized and put into a global context with the help of Stratosphere-Troposphere-Exchange (STE) climatology, trajectory analysis, surface ozone data from other stations, as well as ozone soundings. The analysis shows that El Tololo represents a very remote measuring site, which rarely gets influenced by local pollution, and thus represents an excellent remote GAW station. An indication of this is the relatively small amplitude of the diurnal cycle, even in summer.



The following bullet points summarize the most important conclusions:

•        Only a few data gaps exist in the 20-year-long ozone dataset

•        A positive trend of +0.66 ppb/decade is found up to recent years, which gets weaker from 2010 on and possibly reverses in 2011

•        Over the entire period, the strongest increase in ozone concentrations is observed in austral fall, the strongest decrease in austral summer. The latter is most probably related to different origins of the air masses and greater influence of the Santiago and La Serena region in fall (March-May) than in summer (December-February).

•        In general, the average annual cycle at TLL is dominated by peak concentrations in late spring, followed by a sharp decrease in late winter to early fall, correlating with the shape of average annual STE. TLL can, therefore, be classified as a STE-influenced station, in contrast to stations that are in the marine-boundary-layer or significantly influenced by anthropogenic-pollution.

•        Characterizing the TLL data set with the help of ozone soundings makes it possible to see that the free-tropospheric influence is very strong compared to other stations

•        The maximum ozone concentrations were reached around week 41 (early October) in 1996 and have been retrograding since; recently, maximum concentrations are reached around week 38 (-10 days per decade). This is attributed to a retrogradation in the same magnitude of the maximum in the ST mass flux into the PBL (-21 days per decade) and of the number of STE trajectories around El Tololo (-11 days per decade).

•        The ozone concentrations at El Tololo are ENSO-sensitive. Over the entire year, ozone concentrations are higher during La Niña conditions than during El Niño conditions, especially in late austral spring. This is related to the large-scale atmospheric circulation anomalies over the Pacific, coming along with less NOx-production by lightning and changing circulation patterns.

While many aspects of the O3 time series are well explained, two observations remain unclear and may be elucidated in with the help of regional and a global modelling studies. Firstly, the origin of the retrogradation of the timing of ozone STE maximum is yet unclear. Our hypothesis is that large-scale gravity wave momentum transport has changed over years due to changing tropopause height. Secondly, different ozone trends for different seasons (austral fall versus austral summer) are observed. We postulate that polluted air masses from the greater Santiago area are transported northwards up to La Serena, where the local wind systems (Elqui-Valley-wind) transports the plume up to El Tololo. This process has been confirmed by a short preliminary study with a regional model. At least one year of high-resolution regional model results is required to be able to confirm or reject this hypothesis.

**Acknowledgements**

We acknowledge the support of the Federal Office of Meteorology and Climatology MeteoSwiss through the project Capacity Building and Twinning for Climate Observing Systems (CATCOS) Phase 2, Contract no. 81025332 between the



Swiss Agency for Development and Cooperation (SDC) and MeteoSwiss. Moreover, we would like to express our gratitude towards Michael Sprenger and Bojan Skerlak, who both advised us how to use their ozone STE climatology. As well, we thank Dr. Stephan Henne for his support concerning the FLEXTRA datasets and Dr. Dominik Brunner for his valuable comments. Laura Gallardo is grateful for the support of FONDAP 15110009.

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







**Figure 1: Position of El Tololo, La Serena, Valparaiso and Santiago de Chile including topographic information. Terrain data source: NOAA ETOPO1, plotted using marmap (Pante and Simon-Bouhet, 2013).**



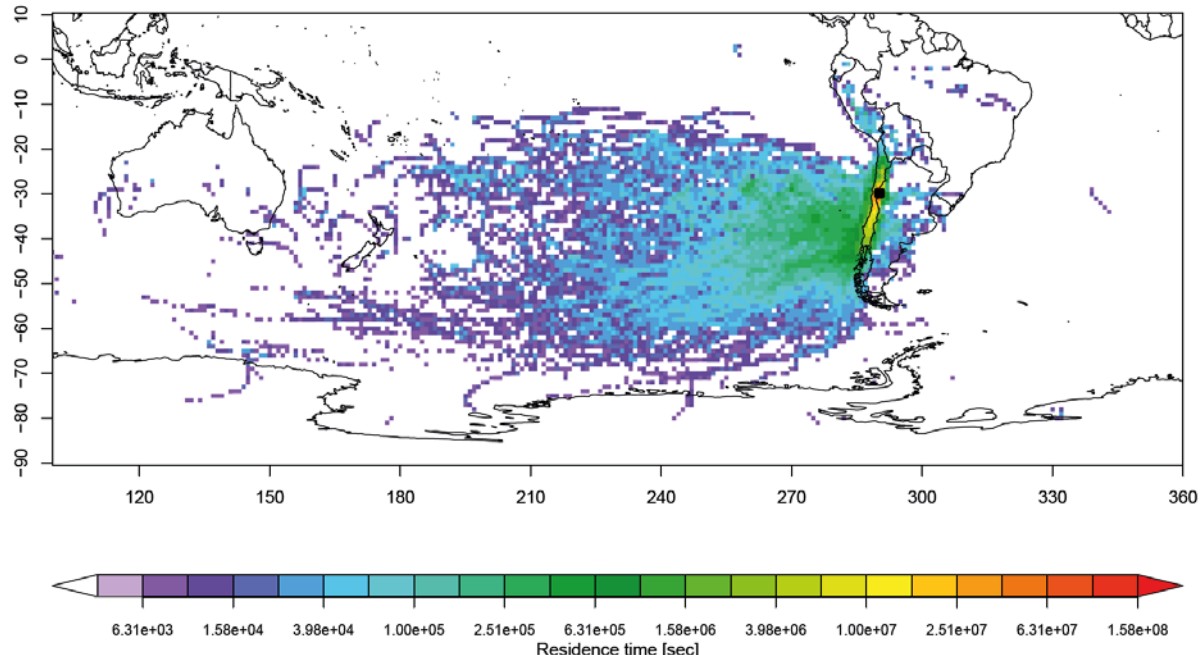

**Figure 2: FLEXTRA trajectory footprint from April 2013- December 2015, origin: TLL, 370m above model topography. Color indicates the total residence time of air parcels, summed up over the time period. TLL is marked with a black dot.**





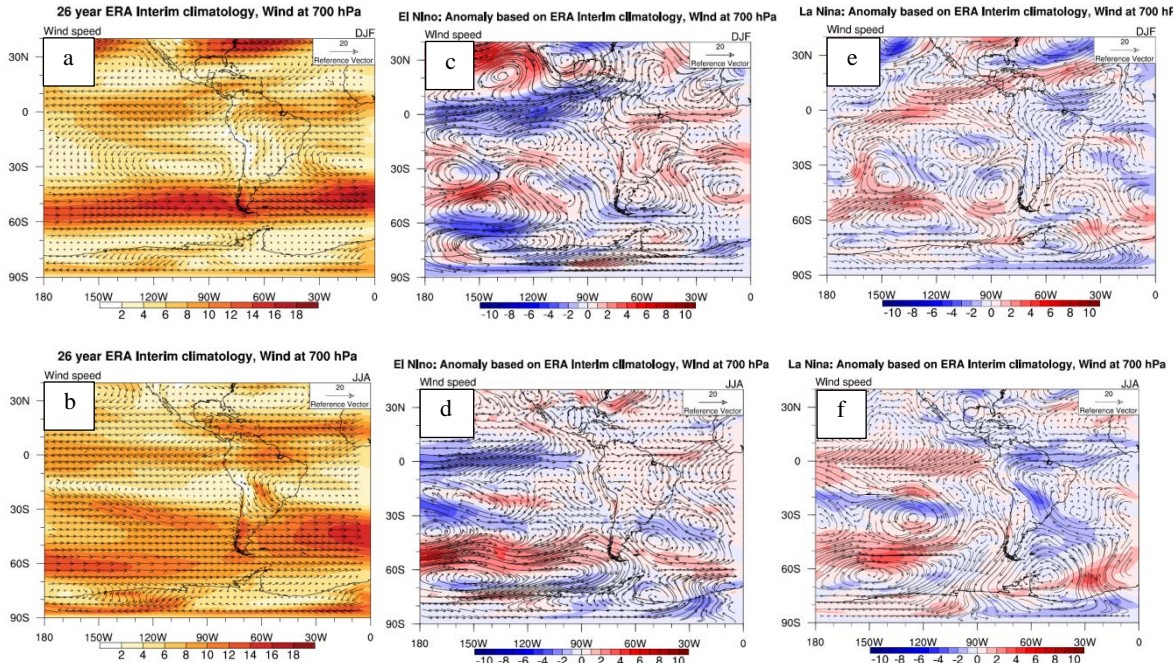

**Figure 3:** ERA Interim wind climatology at 700 hPa (a & b) and wind change in vector and strength during an exemplary El Niño
event (1997-1998) (c & d) and a La Niña event (1988-1989) (e & f).



**Figure 4: Time series of (a) hourly ozone mole fractions at TLL (black) and data gaps (grey); (b) deseasonalized monthly means of ozone mole fractions at TLL (blue) with linear fit (red). Slope of the linear fit is 0.66 ppb/decade; (c) residual trend of the EEMD decomposition.**





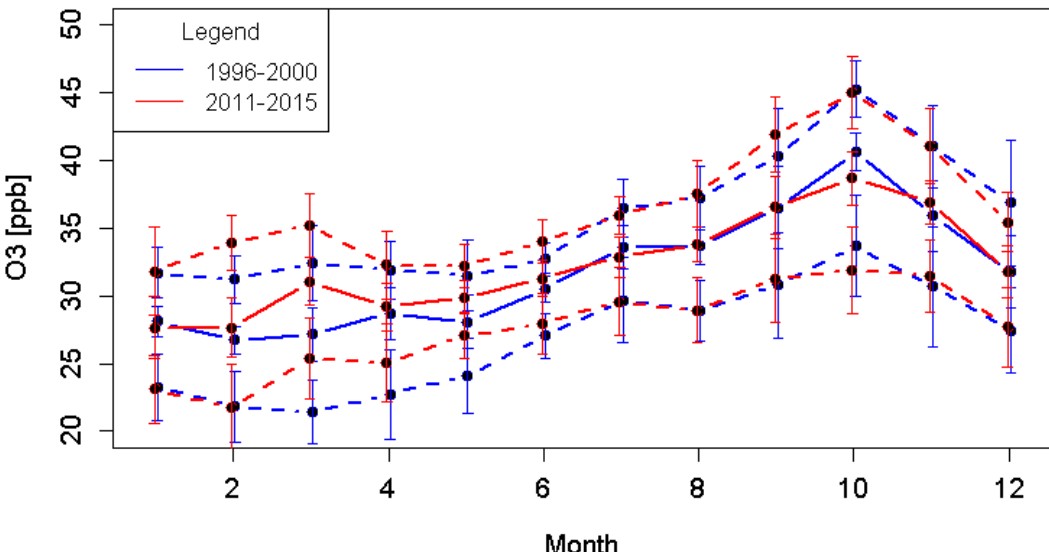

**Figure 5: Mean annual cycles of ozone mole fraction (1996-2000 and 2011-2015) showing mean, upper 95th percentile and lower 5th percentile. For better readability, the monthly means for both periods have been shifted by ±7 days.**





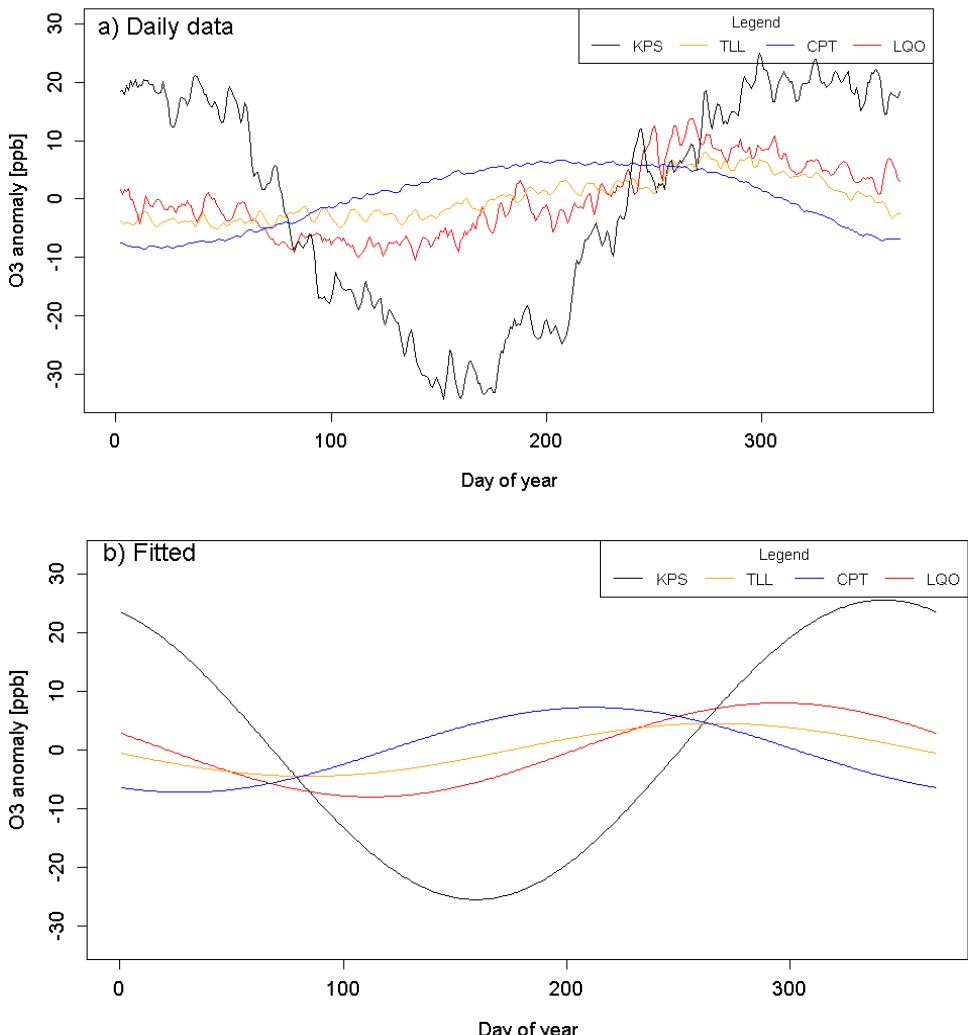

**Figure 7: a) Mean annual cycle of ozone anomalies at different background stations showing a spring maximum (El Tololo [TLL],**
**La Quiaca [LQO], a summer maximum (K-Puszta [KPS]) or a summer minimum (Cape Point [CPT]). Anomalies are deviations**
**from the annual mean. The x-axis shows the day of year. Northern hemispheric data are shifted by 182 days. b) Sine fit to the**
**annual cycles shown in a).**







**Figure 8: Composite plot showing the different southern hemispheric ozone measurements (SHADOZ and WOUDC network and ground-based in-situ data): the x-axis shows the southern latitude, the y-axis represents the delta between the maximum and the minimum of the annual cycle ("Peak-to-peak amplitude"). The size of the points represents the height of the station (ground-based) or of the ozone sonde measurement (SHADOZ & WOUDC). The colors depict the day of year when the maximum of the annual cycle is reached. Points with a black spot illustrate sonde measurements. TLL can be seen at 30°S and an amplitude of 15ppb.**





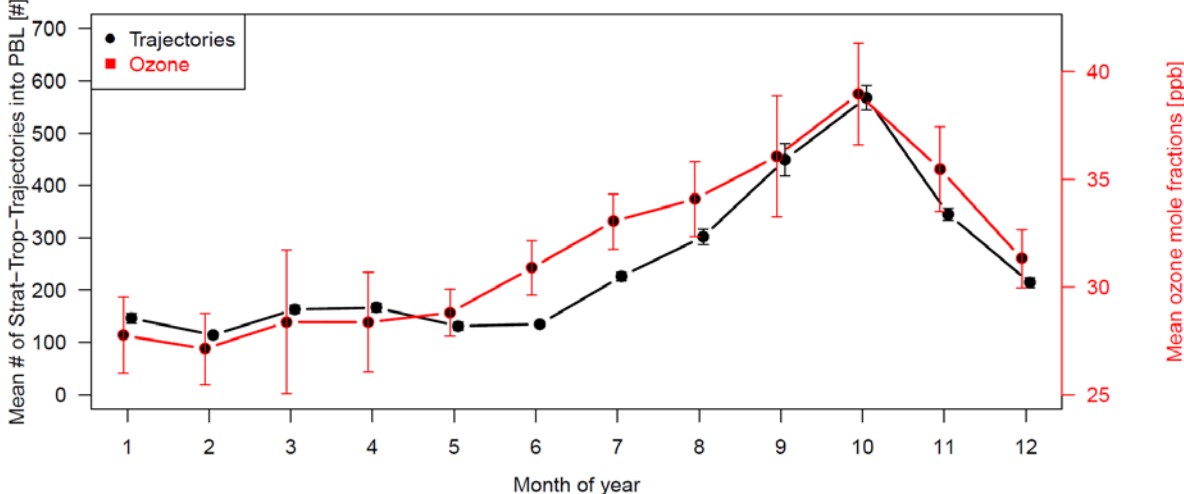

Figure 9: Mean annual cycle over 18 years (1996 - 2015) of ozone at TLL and of trajectories indicating STE above TLL.




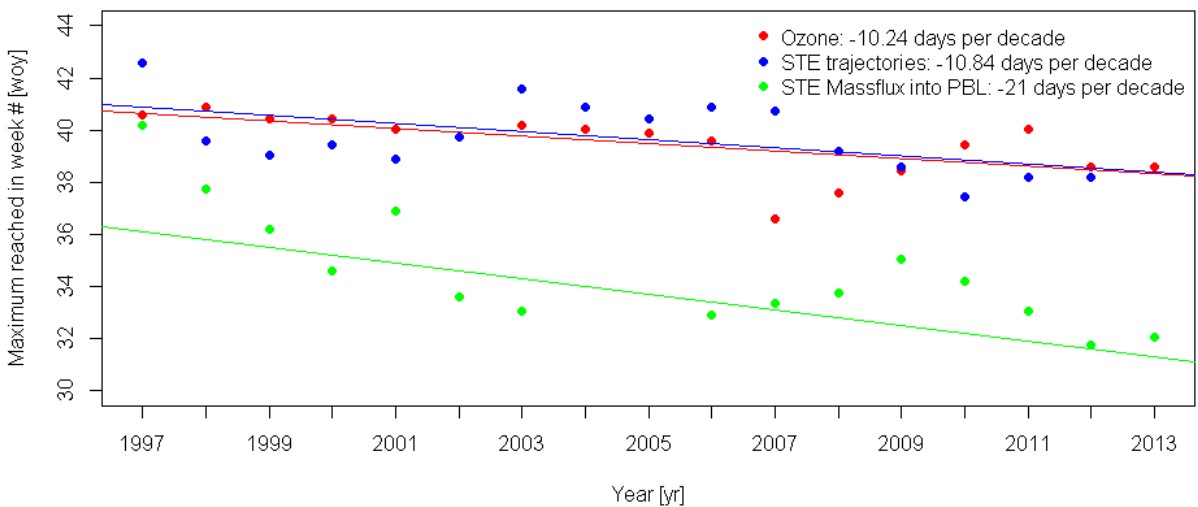

**Figure 10: Time of the year, for which the maximum of ozone (red), STE trajectories (blue), and STE mass flux into the PBL (green) is reached.**





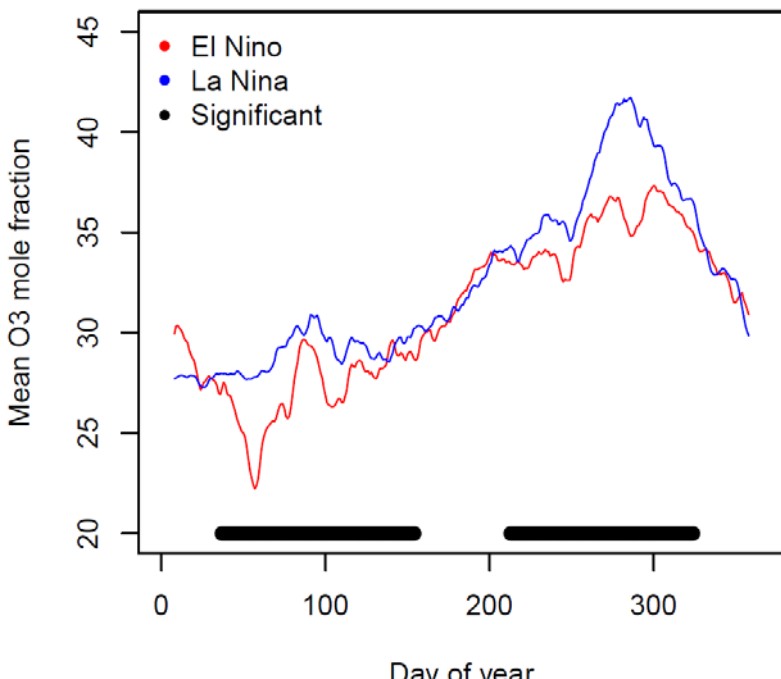

**Figure 11: Computed annual cycle of O₃ mole fractions at TLL in daily resolution during El Niño days (NINO3.4-Index >0.5, total 1914 days between December 1995 and December 2015) and La Niña days (NINO3.4-Index < -0.5, total 2330 days between December 1995 and December 2015). Black point show significant differences between the two curves. NINO3.4 data is derived from daily index reconstructions from SST OI v2 1/4 degree data by NOAA.**



**Table 1: Seasonal linear trends of ozone [ppb / decade] at TLL for a) all-day data (00 to 24 UTC), b) nighttime data from 10 PM to 4 AM, and c) daytime data from 11 AM to 5 PM. Significant trends are labelled with * (95%).**

| Time of day / Season | DJF | MAM | JJA | SON |
|---|---|---|---|---|
| all | -0.2 | 1.6* | 0.2 | 0.1 |
| nighttime | -0.5 | 1.5* | 0.3 | 0.5 |
| daytime | -0.3 | 1.9* | 0.5 | 0.7 |