# Peer review of "Surface ozone in the southern hemisphere: 20 years of data from a site with a unique setting in El Tololo, Chile, 30°N, 71°W, 2200 m asl"

_Atmospheric Chemistry and Physics, 2016_

## Referee Comment (RC1) · Anonymous Referee #1 · 4 Nov 2016

General Comments:

In general the manuscript titled "Surface ozone in the southern hemisphere: 20 years of data from a site with a unique setting in El Tololo, Chile" present a very high quality of science. It is well written and organized.

This study brings a very important contribution to the community about the influence of anthropogenic emission, biomass burning and the stratosphere-to-troposphere transport on tropospheric ozone in the south hemisphere.

The introduction is very clear and allows the reader to really appreciate how a mountain site in the south hemisphere such as El Tololo, Chile is essential for the long-term time

survey of tropospheric ozone. The data and methods are thoroughly detailed and it is very much appreciated. The results, the discussion and the conclusions are clear and easy to read.

Nevertheless, I have some specific comments and technical corrections (see below). Once these corrections are made, I would be favorable for the publication of the manuscript.

Specific Comments:

- Section 1 - Introduction: For the discussion about the seasonal cycle of tropospheric ozone with maximum in spring or in summer Cooper et al., 2014 (section 4) showing seasonal cycle by hemisphere with a maximum in spring for the south hemisphere using OMI/MLS tropospheric column ozone should be cited.

- Section 3.3 - Methods: Is it possible to further justify the 4 ppbv threshold used to exclude high changes between one hour and the next?

- Section 3.3 - Methods: Would the authors confirm that 8.9% represents data influenced by local pollution and missing data? Does it give a first element that ozone at El Tololo is unlikely driven by local pollution?

- Section 3.3 - Methods: Make clear that the trajectories data set are provided by Skerlak et al.

- Section 3.3 - Methods: Would the authors write the definition of the max flux more explicitly as a mathematical formulae? That would be easier for the reader.

- Section 4.1 - Trends: the authors are using Figure 10: Time of the year for which the maximum of zone, STE trajectories and STE mass flux into PBL is reached, to argue in favor of the role of ozone STE max flux in tropospheric ozone seasonal changes and changes in time. This paragraph needs to be clarify. The time periods used to assess the changes of the seasonal cycle of tropospheric ozone over time are 1996-2000 and 2011-2015, whereas the time period in Figure 10 ends in 2013. The shift of ozone

maximum from October to August seems to be seen on Figure 10 but not on Figure 5, why?

I would suggest to add the specific humidity in the study. It can be added on Figure 10 and its seasonal cycle could be shown as well. It will give more evidence of the impact of stratospheric ozone on tropospheric ozone changes.

- Section 4.1 - Trends: The authors are pointing out a shift of two months between maximum of STE flux (Figure A9) and maximum of ozone mixing ratio (Figure 5). This is true for the time period 2011-2015 but not for 1996-2000. In 1996-2000, the STE max flux shows two maxima. This paragraph needs to be clarified.

- Section 4.1 - Trends: Factors influencing the trend in austral fall could be biomass burning in Australia and south of Africa (Cooper et al., 2014), more than Southeast Asia as biomass burning in this region occur mostly in the northern hemisphere.

- Section 4.3 - Large-Scale influences at TLL: As said above, Figure 10 shows time series from 1996 to 2013 and not 2015. Would it be possible to extent the time period to 2015?

- Section 4.3 - Large-Scale influences at TLL: The shift in the seasonal cycle has been discussed in section 4.1 - Trends. I would suggest to move the paragraph of the section 4.1 to the section 4.3, otherwise it is confusing for the reader.

- Section 4.3 - Large-Scale influences at TLL: Instead of discuss the relative humidity, I would suggest to discuss the specific humidity which is the absolute value of humidity in the air.

- Figures 8 and 10: on the y or x-axis, I would suggest to give the month associated to the given week or day of the year. It will help the reader to follow the analysis which is often based on season or month in the text.

- Table 1: I would suggest to add the 95% confidence limit and the p-value

Technical corrections:

- I would suggest to add the longitude and latitude of El Tololo in the title

- L.5 p.3: Change "Unites" to "United"

- L.8 p.7: Change "Stratosphere-Troposphere-Transport" to "Stratosphere-Troposphere-Exchange"

- L.19-20 p.9: I would link both paragraphs (no enter)

- L.22 p.9: Change "positive deviations" to "increase"

- L.23 p.9: Change "During the remaining of the year" to "For the other months of the year"

- L.29 p.9: I would suggest to remove "The attentive reader may have realized that" and start the sentence directly by "The maximum of STE..."

- L.30 p.9: Change "ration" to "ratio"

- L.13 p.10: "biomass burning in ..." a word is missing?

- L.29 to 30 p.: I would suggest to remove this paragraph as everything is already written in the caption of the figure

- Figure 3: I would suggest to change "(a&b)" to "(a) DJF, b) JJA)", same for "(c & d)" and "(e & f)"

- Figure 4: I would suggest to add minor ticks to show all the years

- Figure 5: I would suggest to add minor ticks to show all the months

- Figure A4: I would suggest to change the caption of the figures in order to have the consistent name of the season related to the month: Fall (MAM). It is quite common to order the season as followed: winter, spring, summer, and fall

- Figure A6: I would suggest to follow the common order: winter (JJA), spring (SON),

summer (DJF), and fall (MAM)

- Figure A7: I would suggest to add dash lines for the three panels as in Figure 4 and add minor ticks to show all the years

- Figure A9: I would suggest to add minor ticks to show all the month

- In general be careful with the use of "STE" and "STT". STE = stratosphere - troposphere exchange, STT = stratosphere to troposphere transport
* * *

---

## Referee Comment (RC2) · Anonymous Referee #2 · 9 Dec 2016

**A. Major comments**

The authors present a 20-year time series of lower tropospheric ozone measurements at a high-elevation site in South America and discuss the influences of ENSO and STE on the observed ozone seasonal cycle and long-term trends. This long-term ozone observational record is extremely valuable because observations are very sparse in the Southern Hemisphere, as the authors noted. The manuscript is valuable in this perspective. However, I cannot recommend publishing this manuscript in its present form because discussions on the processes controlling ozone at this site in the current manuscript are somewhat unclear, inconsistent or incomplete:

1) According to Figure 5, ozone measured observed at El Tololo increases substantially

in the austral autumn (March-April) but shows some decreases in October. The largest ozone differences between El Nino and La Nina years also appear in March-April and September-October (Figure 11). However, the meteorological fields in Figure 3 are shown for DJF and JJA, which are not relevant to the key seasonal features shown in Figures 5 and 11.

2) Figure 4b shows a time series for the deseasonalized monthly ozone data. Based on the plot, the authors noted in the text that it is not clear to see an ENSO signal. However, the influence of ENSO on ozone is known to have a strong seasonality (see also Figure 11). Why not also show a time series of monthly ozone in March-April and September-October, respectively, and correlate the time series with the ENSO index? Should the slope shown on the top of Fig.4b be ppb/decade rather than ppb/y? It would be nice to also report the 95% confidence limits of the trends for annual mean and for each season.

3) Figure 7 is very confusing. The figure caption notes that data at some sites are shifted by 182 days. Can you just separate the plot for the sites in the Northern Hemisphere versus the Southern Hemisphere without shifting the days and clearly label the latitude and longitude of each site? Another possible factor contributing to the seasonal cycle of ozone in the southern Hemisphere is biomass burning emissions, which the authors did not discuss at all. Please check the seasonal cycle of biomass burning activity in this region as reported in the published literature.

**B. Recommendations**

Similar to Mauna Loa Observatory in the Northern Hemisphere (Lin et al., 2014,Nature Geoscience), El Tololo ( 30S) is located in the subsiding branch of the Hadley Cell in the Southern Hemisphere. Ozone measured at Mauna Loa increases during boreal autumn (Sep-Oct) but shows no significant trend during boreal spring (March-April). Interestingly, ozone measured at El Tololo shows an increase during austral autumn (March-April) but no trend in austral spring (Sep-Oct). While the mechanisms controlling ozone trends at these two sites may be different, there are some similarities on their seasonal ozone trends. Thus, the referee strongly encourages the authors to carefully read Lin et al. (2014) and organize the analyses and associated discussions for El Tololo in a similar way to Mauna Loa.

Lin, M.Y., L.W. Horowitz, S. J. Oltmans, A. M. Fiore, Songmiao Fan (2014): Tropospheric ozone trends at Manna Loa Observatory tied to decadal climate variability, Nature Geoscience, 7, 136-143, doi:10.1038/NGEO2066.

**The referee recommends the authors reorganize the figures as follows**:

**Figure 1**: Position of El Tololo.

**Figure 2**: Seasonal ozone trends observed at El Tololo (Figure 5 in the current manuscript)

**Figure 3**: 500 hPa wind speed and mean sea level pressure for four representative months: Jan, March, July, October. Label the location of El Tololo on the maps.

Here you can discuss the position of El Tololo relative to the mean jet location and associated sources of ozone variability (Similar to Fig.2 in Lin et al. (2014) and associated discussions therein)

**Figure 4**: Trajectory footprints (similar to Figure 2 in the current manuscript) but separately for the four representative months

**Figure 5**: Time series for deseasonalized, March-April, and September-October monthly ozone, along with the ENSO index

**Figure 6**: Figure 11 in the current manuscript

**Figure 7**: Additional analysis for the decadal changes in 500 hPa winds and geopogential heights between the two periods when you see a shift in ozone trend based on the analysis in Figure 5.

[Figure]

Please see also Fig.4 in the following manuscript. Their model also shows that free tropospheric ozone near El Tololo increases during austral autumn (MAM) but there is no significant trend in austral winter (JJA). I wonder if the observed ozone increase at El Tololo has something to do with the poleward shift of the subtropical jet stream in the Southern Hemisphere or changes in geopotential height patterns. The analysis suggested above will help in interpreting the cause of the seasonal trends.

Lin, M.Y., W. Horowitz, R. Payton, A.M. Fiore, G. Tonnesen. US surface ozone trends and extremes over 1980-2014: Quantifying the roles of rising Asian emissions, domestic controls, wildfires, and climate. Atmos. Chem. Phys. Discuss., doi:10.5194/acp-2016-1093, accessible at http://www.atmos-chem-phys-discuss.net/acp-2016-1093/

**C. Additional suggestions**

Reading the following articles might be useful for understanding the connections between ENSO and ozone variability. Introduction in the current manuscript focuses too much on precursor emission changes that are most relevant for polluted regions in the Northern Hemisphere. I think citing and discussing the findings from the following papers are more relevant to your analysis.

(1) ENSO, changes in deep convection over the tropics and associated ozone variability

Doherty, R. M., D. S. Stevenson, C. E. Johnson, W. J. Collins, and M. G. Sanderson (2006), Tropospheric ozone and El Nino-Southern Oscillation: Influence of atmospheric dynamics, biomass burning emissions, and future climate change, J. Geophys. Res., 111 (D19), D19304, doi: 10.1029/2005jd006849.

Ziemke, J. R., Chandra, S., Oman, L. D., and Bhartia, P. K.: A new ENSO index derived from satellite measurements of column ozone, Atmos. Chem. Phys., 10, 3711-3721, doi:10.5194/acp-10-3711-2010, 2010.

(2) ENSO, STE and ozone variability at northern mid-latitudes:

Lin, M.Y., A.M. Fiore, L.W. Horowitz, A.O. Langford, S. J. Oltmans, D. Tarasick,

H.E. Reider (2015): Climate variability modulates western US ozone air quality in spring via deep stratospheric intrusions, Nature Communications, 6, 7105, doi:10.1038/ncomms8105

---

## Author Comment (AC1) · 4 Feb 2017

**acp-2016-617: Review #1**

*"Surface ozone in the southern hemisphere: 20 years of data from a site with a unique setting in El Tololo, Chile*" by J. G. Anet et al.

We would like to thank the anonymous referee #1 for his/her great review of our publication. Below, we provide the answers to his/her comments.

**Section 1 - Introduction: For the discussion about the seasonal cycle of tropospheric ozone with maximum in spring or in summer Cooper et al., 2014 (section 4) showing seasonal cycle by hemisphere with a maximum in spring for the south hemisphere using OMI/MLS tropospheric column ozone should be cited.**

We added following sentence in the "introduction", Page 3, L20: "*These findings were more broadly confirmed by Cooper et al. (2014), who, using satellite-measured total column ozone datasets, classified the onsets of the Total Column Ozone (TCO) maxima globally. In general, spring TCO maxima are found rather on the SH, while summer TCO maxima are prominent in the NH.*"

**Section 3.3 - Methods: Is it possible to further justify the 4 ppbv threshold used to exclude high changes between one hour and the next?**

The 4 ppb limit has been defined after thorough trial-and-error experiments. Those 4 ppb created least "false negatives" or "false positives" during automatic filtering, which otherwise would have to be corrected manually. This value changes from station to station, as it depends from the natural ozone variability. We added the following to the paper: "*This value of 4 ppb has been defined as such to avoid too many false positives or negatives during the automatic filtering process, in order to minimize the workload during the manual dataset review process.*"

**Section 3.3 - Methods: Would the authors confirm that 8.9% represents data influenced by local pollution and missing data? Does it give a first element that ozone at El Tololo is unlikely driven by local pollution?**

Thank you for this valuable comment. The reviewer is right that the 8.9% do cover both the excluded data by the filtering process as well as missing data. This number is dominated by the larger data gaps (see grey periods in Fig. 4 and Table S1) and thus, does not give neither a meaningful impression of the rigor of the filter nor of the frequency of rare events of local pollution. The filtering process only excludes 4.9% from the available data. This information will be added to the manuscript. Indeed, a fraction of only 4.9% being identified as influenced by local pollution is a first benchmark indicating the pristine setting of the station. It is worth to mention that it is generally difficult to unambiguously

classify data points as "local pollution events" or "regional pollution events" as periods with steadily elevated mole fractions would not be excluded by the filter. However, the absence of frequent local emissions (e.g. due to the operation of the back-up diesel generator or heavy traffic on the Cerro Tololo premises) is also in line with the observations of the station operators. Therefore, we rephrased the respective sentence that it reads: "*The filtering of the data excludes approximately 4.9% of the available data indicating the pristine setting of the sampling site with hardly any influence from local pollution sources from the premises' infrastructure.*"

**Section 3.3 - Methods: Make clear that the trajectories data set are provided by Skerlak et al.**

In order to avoid any misunderstanding, following sentence was reformulated: "*Driven by the wind field of ERAI, Škerlak et al. (2014) calculated kinematic trajectories using an 3-steps iterative Eulerian integration scheme (Sprenger and Wernli, 2015).*"

**Section 3.3 - Methods: Would the authors write the definition of the max flux more explicitly as a mathematical formulae? That would be easier for the reader.**

Thanks for this good suggestion. We adapted the manuscript accordingly: $\Delta MF_{O_3} \approx n * t * \Delta m_{O_3}$

**Section 4.1 - Trends: the authors are using Figure 10 [...] to argue in favor of the role of ozone STE max flux in tropospheric ozone seasonal changes and changes in time. This paragraph needs to be clarified. The time periods used to assess the changes of the seasonal cycle of tropospheric ozone over time are 1996-2000 and 2011-2015, whereas the time period in Figure 10 ends in 2013. The shift of ozone maximum from October to August seems to be seen on Figure 10 but not on Figure 5, why?**
**I would suggest to add the specific humidity in the study. It can be added on Figure 10 and its seasonal cycle could be shown as well. It will give more evidence of the impact of stratospheric ozone on tropospheric ozone changes.**

The reviewer's interpretation is totally comprehensible and we would like to clarify certain points. First, the period shown in Fig. 10 runs only from 1997 to 2013, i.e. no data are shown for 1995, 1996, 2014 and 2015, due to the way the analysis routine works. The intrinsic mode functions (IMF) was calculated for 4 year sliding windows, therefore "cutting" the first and last two years. Calculating the IMF over yearly data would lead to less meaningful/representative results due to the relatively low signal-to-noise-ratio (interannual variability). The shift of the timing in the maximum of ozone is less visible in Figure 5 also due to the different data treatment. Figure 5 shows a comparison of 3 different percentiles of 5-year-monthly averages, not absolute maximal values, and data are aggregated in monthly bins. We appreciate the suggestion to use specific humidity. Usually, ozone peaks associated with stratospheric intrusions are mostly accompanied with low relative humidity (RH) as it was also

shown by Rondanelli et al. (2002) for El Tololo. We also looked into the RH measurements. Unfortunately, the RH time series at El Tololo suffers from sensor deterioration which jeopardizes a detailed inclusion of the RH data into the analysis (see Fig. S7 c).

In order to help the reader to interpret the Figures, we changed the manuscript, the caption of Figure 9 and the text (see Section 4.3 of the revised manuscript) as following: "*For calculation, a 4-year sliding window of daily data was defined and run over all data between 1996 and 2015.*"

And: "*Note that the regression is only poorly visible in Fig. 5, in which data are aggregated in monthly bins and a comparison of 3 different percentiles of 5-year-monthly averages, instead of absolute maximal values, is shown.*"

And: "*A 4-year sliding window of daily data was applied. Average values for the years 1996-1999 are shown as the data point in end of December 1997, 1997-2000 in end of December 1998, and so forth until years 2012-2015 which are shown as data point in end of December 2013.*" was added to the caption of Figure 9.

**Section 4.1 - Trends: The authors are pointing out a shift of two months between maximum of STE flux (Figure A9) and maximum of ozone mixing ratio (Figure 5). This is true for the time period 2011-2015 but not for 1996-2000. In 1996-2000, the STE max flux shows two maxima. This paragraph needs to be clarified.**

Thanks for this remark. It is true that the STE trajectory frequency of the earlier period shows two maxima. We erroneously focused purely on the more recent period. The finding of the two maxima (or rather: the persistent low in August) in the 1996-2000 period is robust and can be found in the absolute maxima, mean, median and 5-95 percentiles. The reason for this finding is not entirely clear – we assume that the state of the QBO and ENSO might have played an important role on STE during this period (relatively strong El Niño, weaker la Niña), as shown in Neu et al. (2014). As can be seen in Fig. 1R1 and Fig. 2R1 (below), the QBO shear index (QBO, red lines) and the multivariate ENSO index (MEI, blue lines) where in-phase during nearly two years between 1996-2000 and relatively out-of-phase during 2011-2015. The effects are known: the tropical upwelling increases, boosting the planetary-scale wave activity and henceforth the stratospheric circulation, finally leading to an increase in STE exchange in the sub- and extratropics (Neu et al. 2014 & references therein). This most likely leads to an overall stronger STE activity especially in JJA 1997 and JJA 1998. Also due to the comments of reviewer #2, we have entirely reformulated section 4.3

[Figure]

**Fig. 1R1: ENSO MEI (blue) and QBO shear index (red) for the 1996-2000 period.**

[Figure]

**Fig. 2R1: ENSO MEI (blue) and QBO shear index (red) for the 2011-2015 period**

**Section 4.1 - Trends: Factors influencing the trend in austral fall could be biomass burning in Australia and south of Africa (Cooper et al., 2014), more than Southeast Asia as biomass burning in this region occur mostly in the northern hemisphere.**

This is correct. However, by austral fall, the Southeast-Asian branch of the ITCZ has already moved northwards, allowing first pollutants to be transported southbound of the ITCZ into the southern hemisphere. Moreover, biomass burning emissions in Australia and Southern Hemisphere Africa are minimal in austral fall (van der Werf et al., 2006). For clarity, we rewrote the following in Section 4.3 of the revised manuscript: *"An increase of biomass burning in Southeast Asia (e.g. Shi and Yamaguchi, 2014;Verma et al., 2015) and Australia (Cooper et al., 2014) with subsequent eastward transport of ozone precursors, could also explain the positive anomaly in MAM in the 2011-2015 period, as the Northward*

*migration of the ITCZ during this time of the year starts to allow effects of NH emissions to be seen in the SH and prevailing westerly conditions (see Fig. 2) exclude any sensitivity of ozone mole fractions at TLL to emissions on the South American continent."*

**Section 4.3 - Large-Scale influences at TLL: As said above, Figure 10 shows time series from 1996 to 2013 and not 2015. Would it be possible to extent the time period to 2015?**

As replied above, this is technically not possible.

**Section 4.3 - Large-Scale influences at TLL: The shift in the seasonal cycle has been discussed in section 4.1 - Trends. I would suggest to move the paragraph of the section 4.1 to the section 4.3, otherwise it is confusing for the reader.**

This is a very good idea. We moved paragraph "seasonal cycles" in 4.1. to 4.3 and, together with the remarks of reviewer #2, restructured and reformulated the section extensively.

**Section 4.3 - Large-Scale influences at TLL: Instead of discuss the relative humidity, I would suggest to discuss the specific humidity which is the absolute value of humidity in the air.**

As said above, we could only base ourselves on some very short RH measurements done at TLL, as the sensors mostly were reporting wrong values. Since 2014, the station has been equipped with new sensors, and we totally agree that such a study, relating tropospheric ozone with specific humidity, should absolutely be carried out in near future.

**Figures 8 and 10: on the y or x-axis, I would suggest to give the month associated to the given week or day of the year. It will help the reader to follow the analysis which is often based on season or month in the text.**

We agree that the reader could be helped by completing the y-axis of Figure 10 with months: We have therefore adapted Figure 10 as the reviewer suggested:

[Figure]

**Fig. 3R1: Revised Fig. 10 of original manuscript**

**Table 1: I would suggest to add the 95% confidence limit and the p-value**

We added the 95% confidence limit and the p-value in Table 1 as suggested. On the same time, we slightly revised the numbers as we updated the analysis routine. The latter finding nevertheless does not influence the overall trend discussion.

**Typos/Technical corrections**

**I would suggest to add the longitude and latitude of El Tololo in the title**

We agree this would be a good idea. We therefore revised the title: *Surface ozone in the southern hemisphere: 20 years of data from a site with a unique setting in El Tololo, Chile, 30°N, 71°W, 2200 m asl*

**L.5 p.3: Change "Unites" to "United"**

**L.8 p.7: Change "Stratosphere-Troposphere-Transport" to "Stratosphere Troposphere-Exchange"**

**L.19-20 p.9: I would link both paragraphs (no enter)**

**L.22 p.9: Change "positive deviations" to "increase"**

**L.23 p.9: Change "During the remaining of the year" to "For the other months of the year"**

We adopted the suggested changes.

**L.29 p.9: I would suggest to remove "The attentive reader may have realized that" and start the sentence directly by "The maximum of STE..."**

We agree that this makes the text more readable and adapted the manuscript.

**L.30 p.9: Change "ration" to "ratio" - L.13 p.10: "biomass burning in ..." a word is missing?**

We corrected the typos

**L.29 to 30 p.: I would suggest to remove this paragraph as everything is already written in the caption of the figure**

We assumed that the reviewer meant L29-30 p11, and we have removed the double information from the text about Fig. 8.

**Figure 3: I would suggest to change "(a&b)" to "(a) DJF, b) JJA)", same for "(c & d)" and "(e & f)"**

Good idea, we changed the figure caption accordingly.

**Figure 4: I would suggest to add minor ticks to show all the years - Figure 5: I would suggest to add minor ticks to show all the months**

We modified the figures, adding the minor ticks:

[Figure]

Fig. 4R1: Revised Fig. 4 of original manuscript

[Figure]

Fig. 5R1: Revised Fig. 5 of original manuscript

**Figure A4: I would suggest to change the caption of the figures in order to have the consistent name of the season related to the month: Fall (MAM). It is quite common to order the season as followed: winter, spring, summer, and fall**

We modified the figure caption accordingly, as well as the order:

[Figure]

**Fig. 6R1: Revised Fig. A4 of original manuscript**

**Figure A6: I would suggest to follow the common order: winter (JJA), spring (SON), summer (DJF), and fall (MAM)**

We reordered the seasons following the suggestion of the reviewer.

**Figure A7: I would suggest to add dash lines for the three panels as in Figure 4 and add minor ticks to show all the years**

We adapted the minor ticks on the x-axis as suggested:

[Figure]

Fig. 7R1: Revised Fig. A7 of original manuscript

**Figure A9: I would suggest to add minor ticks to show all the month - In general be careful with the use of "STE" and "STT". STE = stratosphere - troposphere exchange, STT = stratosphere to troposphere transport**

Thanks for pointing out the wrong naming. We modified the figure caption and added minor ticks:

[Figure]

**Fig. 8R1: Revised Fig. A9 of original manuscript; Caption modified to:** *Mean annual ozone STE mass flux cycle (1995-2000 and 2010-2015) showing mean, upper 95th percentile and lower 5th percentile*

**References**

Neu, J. L., Flury, T., Manney, G. L., Santee, M. L., Livesey, N. J., and Worden, J.: Tropospheric ozone variations governed by changes in stratospheric circulation, Nature Geosci, 7, 340-344, 10.1038/ngeo2138

Rondanelli, R., Gallardo, L., and Garreaud, R. D.: Rapid changes in ozone mixing ratios at Cerro Tololo (30°10′S, 70°48′W, 2200 m) in connection with cutoff lows and deep troughs, Journal of Geophysical Research: Atmospheres, 107, 1-15, 10.1029/2001JD001334, 2002.

van der Werf, G.R., Randerson, J. T., Giglio, L., Collatz, G. J., Kasibhatla, P. S., and Arellano, A. F. Jr, Interannual variability in global biomass burning emissions from 1997 to 2004, Atmos. Chem. Phys., 6, 3423–3441, 2006

---

## Author Comment (AC2) · 4 Feb 2017

**acp-2016-617: Review #2**

*"Surface ozone in the southern hemisphere: 20 years of data from a site with a unique setting in El Tololo, Chile*" by J. G. Anet et al.

We would like to thank the anonymous referee #2 for his/her critical review of our publication. Below, we provide the answers to his/her comments.

**Major comments of the reviewer**

**1) According to Figure 5, ozone measured observed at El Tololo increases substantially in the austral autumn (March-April) but shows some decreases in October. The largest ozone differences between El Nino and La Nina years also appear in March-April and September-October (Figure 11). However, the meteorological fields in Figure 3 are shown for DJF and JJA, which are not relevant to the key seasonal features shown in Figures 5 and 11.**

Thanks for pointing this out. It is true that the climatology, as it is shown now, does not create any benefit for the reader. We therefore decided to show MAM and SON instead of DJF and MAM in Fig. 3 of the revised manuscript:

[Figure]

**Fig. 1R2: ERA Interim wind climatology at 700 hPa (a, MAM and b, SON) and wind change in vector and strength during an exemplary El Niño event (1997-1998) (c, MAM and d, SON) and a La Niña event (1988-1989) (e, MAM and f, SON).**

**2) Figure 4b shows a time series for the deseasonalized monthly ozone data. Based on the plot, the authors noted in the text that it is not clear to see an ENSO signal. However, the influence of ENSO on ozone is known to have a strong seasonality (see also Figure 11). Why not also show a**

**time series of monthly ozone in March-April and September-October, respectively, and correlate the time series with the ENSO index?**

Thanks for this valuable critique. It is true that the entire section 4.3 needs a thoughtful restructuring. Based on your suggestion, we tried to include a more in-depth discussion of the ENSO signal. We also included your idea to correlate the ozone time series with the MEI for the two periods (March-April and September-October) and have modified sect. 4.3 accordingly. We refer to "recommendations of the reviewer" for a more complete reply.

**Should the slope shown on the top of Fig.4b be ppb/decade rather than ppb/y?**

This is correct. We modified the slope legend.

**It would be nice to also report the 95% confidence limits of the trends for annual mean and for each season.**

We added, as also recommended by reviewer #1, the p-value and the confidence interval. Significant trends with 90% and 95% confidence are now labelled with * and **, respectively.

**3) Figure 7 is very confusing. The figure caption notes that data at some sites are shifted by 182 days. Can you just separate the plot for the sites in the Northern Hemisphere versus the Southern Hemisphere without shifting the days and clearly label the latitude and longitude of each site?**

Thanks for this comment. We agree that the shifting by half a year may be surprising to the reader, but makes the comparison of stations on the Northern hemisphere and the Southern hemisphere easier, as the seasons on both hemispheres will be aligned over another. In order to increase the readability, we modified the x-axis labels to name the season, and no more the day of year. We prefer this version rather than a separation of the plots, as this would result in a total of 4 plots, without significantly increasing the clarity. Moreover, we decided to add Mauna Loa (MLO) to the plot in order to include the station in the entire discussion.

[Figure]

[Figure]

**Another possible factor contributing to the seasonal cycle of ozone in the southern Hemisphere is biomass burning emissions, which the authors did not discuss at all. Please check the seasonal cycle of biomass burning activity in this region as reported in the published literature.**

We are thankful that the reviewer pointed out this topic. It is however not entirely true that we did not discuss biomass burning at all. Several times, biomass burning is shortly discussed as a possible factor. Different studies have been cited concerning this topic. We can only repeat that regional biomass burning in the El Tololo region as well as over most of the Southern American continent is not a factor considering the footprint of air parcels travelling to El Tololo, and that influences from Africa or Australia are rather improbable (van der Werf et al., 2006). We nevertheless considered this point when editing section 3.4.

**Recommendations of the reviewer**

**Similar to Mauna Loa Observatory in the Northern Hemisphere (Lin et al., 2014,Nature Geoscience), El Tololo ( 30S) is located in the subsiding branch of the Hadley Cell in the Southern Hemisphere. Ozone measured at Mauna Loa increases during boreal autumn (Sep-Oct) but shows no significant trend during boreal spring (March-April). Interestingly, ozone measured at El Tololo shows an increase during austral autumn (March-April) but no trend in austral spring (Sep-Oct). While the mechanisms controlling ozone trends at these two sites may be different, there are some similarities on their seasonal ozone trends. Thus, the referee strongly encourages the authors to carefully read Lin et al. (2014) and organize the analyses and associated discussions for El Tololo in a similar way to Mauna Loa. (...*following suggestion of restructuring the order of the figures*...)**

We are grateful that the reviewer makes the link to Mauna Loa, another remote, high-altitude station located on the Northern Hemisphere. We did not discuss Mauna Loa due to different reasons:

- This paper was meant to present and discuss 20 years of ozone data from a southern hemispheric station, additionally to put it in context with other stations on both hemispheres, but not to compare it with one similar station from the northern hemisphere
- It is closer to upstream regions with large anthropogenic activities and thus, most likely stronger influenced by ozone precursors emissions
- Its location, more than 1000 km closer to the Equator, makes it difficult to classify it as a non-equatorial station

Yet, we realize that this was possibly a bit short-sighted. As noted above, we restructured section 4.3 entirely, also including a short discussion of similarities and differences to Mauna Loa, including the implicated MEI and QBO shear indexes. However, we tend to disagree that we should organize the analyses similar to the paper of Lin et al. (2014) due to following reasons:

- The manuscript intends to present a new station to the scientific community, first explaining its "special setting" compared to other GAW stations on the world
- We attempt to start from straightforward, rather simple-to-understand connections (like a simple trend analysis) and to later focus into deeper insights, analyzing annual cycles, comparing the station to other stations as well as ozone sonde measurements (in order to get a 3D-picture) and finally trying to link the recognitions with large-scale interactions like ENSO, or STE
- We followed the approaches of recent multi-station or multi-platform (in-situ, but also soundings, airborne observations, satellite products) O3 trend analysis (Logan et al., 2012;Parrish et al., 2013, 2012;Cooper et al., 2014) where changes in emissions and ozone photochemistry, changes in transport and pollution transport pathways, and potential of influence of climate change are investigated side-by-side

**Please see also Fig.4 in the following manuscript. Their model also shows that free tropospheric ozone near El Tololo increases during austral autumn (MAM) but there is no significant trend in austral winter (JJA). I wonder if the observed ozone increase at El Tololo has something to do with the poleward shift of the subtropical jet stream in the Southern Hemisphere or changes in geopotential height patterns. The analysis suggested above will help in interpreting the cause of the seasonal trends.**

We have attentively read the manuscript of Lin et al. (2016) and included a short discussion about the subtropical jet stream into section 4.3.

**Reading the following articles might be useful for understanding the connections between ENSO and ozone variability. Introduction in the current manuscript focuses too much on precursor emission changes that are most relevant for polluted regions in the Northern Hemisphere. I think citing and discussing the findings from the following papers are more relevant to your analysis.**

Thanks for these recommendations. They were of great help when restructuring section 4.3. We also included some ENSO-related introductions, citing the suggested publications and others:

[revised manuscript text omitted]

---

## Author Response (AR2)

Dear Dr. West

Many thanks for your message concerning the "minor revisions" of our paper, although reviewer #2 would like to reject our paper.

We can answer the comments of referee #2 as following:

**1.** The largest ozone increase occur in March and April, with little change in May. Why do you average the wind patterns over March-April-May?**

It is true that the largest ozone increase occurs in the months of March and April. Nevertheless, the common approach is to split the year – and especially climatologies – into seasonal means (December January February / March April May / June July August / September October November). For the sake of comparability, we therefore also included the month of May.

We nevertheless generated the plots for March-April and September-October (the latter are the two months with the strongest rise in ozone in the annual cycle). When comparing Figure RR1 below with Fig. 3 of the resubmitted version, only minor differences can be seen which do not change the conclusions drawn. Therefore, we prefer sticking to the common approach if possible.

---

## Author Response (AR3)

Dear Dr. West

Many thanks for the fantastic collaboration we had with you over the past 5 months.

We have adopted your suggestion by removing the coordinates in the title – which were obviously wrong anyway...

Kind regards

Julien Anet and all co-authors

[revised manuscript text omitted]